# Soft Skills and Study-Related Factors: Direct and Indirect Associations with Academic Achievement and General Distress in University Students

Nicole Casali *[image ORCID] and Chiara Meneghetti [image ORCID]

Department of General Psychology, University of Padova, 35131 Padova, Italy; chiara.meneghetti@unipd.it
* Correspondence: nicole.casali@phd.unipd.it

**Abstract:** Numerous noncognitive factors have been shown to influence students' academic and nonacademic outcomes, yet few studies have contemporarily studied these factors to understand their specific roles. The present study tested a model in which five soft skills (i.e., epistemic curiosity, creativity, critical thinking, perseverance, and social awareness) were conceived as personal qualities that influence achievement and general distress through the mediation of four study-related factors (i.e., achievement emotions, self-regulated learning strategies, motivational beliefs, and study resilience). A total of 606 Italian university students (153 males, $M_{age}$ = 22.74, $SD_{age}$ = 3.72) participated in the study and completed self-report measures of soft skills, study-related factors, and general distress measures; grades were considered for academic achievement. Results showed that all four study-related factors significantly mediated the relationship of soft skills with academic achievement, while only achievement emotions and study resilience emerged as significant mediators between soft skills and general distress. Our findings indicated complex relations between individual factors and students' outcomes due to several factors. The theoretical and practical implications are discussed.

**Keywords:** soft skills; academic achievement; mental health; self-regulated learning; emotions

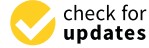



## 1. Introduction

What makes students successful has puzzled scholars and practitioners for decades [1]. Indeed, when considering higher education students, a host of cognitive and noncognitive factors come into play and have been found to significantly affect relevant outcomes, such as academic achievement, which is usually operationalized as GPA or grades (see Richardson et al., [2] for a meta-analytical account), or mental health symptoms, which are generally referring to depression, anxiety, and stress symptoms [3,4], that could help define what success looks like in this population. Some of these two outcomes' relevant identified correlates include several individual so-called soft skills (e.g., emotional intelligence; [5,6]) and study-related factors (e.g., achievement emotions, self-regulated learning, and motivation to learn; [2,7,8]). However, what happens when these factors are considered together? Do they help explain distinct portions of variance? Further, what are their reciprocal relationships? Here, we propose a model considering the direct and indirect relationships possibly linking several soft skills and study-related factors with academic achievement and psychological distress in a large sample of university students.

## 2. Soft Skills and Students' Outcomes

Soft skills [9,10] is an umbrella term used in the literature to encompass a series of acquirable personal qualities and competencies that seem to increasingly be required to face the challenges the labor market poses [11]. Following the framework proposed by the World Economic Forum [11], here we consider the following soft skills (see Table 1 for an overview): epistemic curiosity (desire to know), creativity (originality), critical thinking (analyzing and questioning), perseverance (prolonged effort to achieve goals), and social

awareness (efficient interpersonal interaction). These personal qualities help individuals regulate their emotions, cognitions, and behaviors to fulfil their personal goals [12,13]. As single constructs, they have received quite a bit of attention in the literature with regard to their associations with academic achievement [2,5,14–16](. The identified relationships are usually small (around $r = 0.10$) and may indicate that these kinds of features could exert a greater influence on academic achievement through the mediation of more circumscribed, study-related factors (as noted by previous studies: [17,18]. As for mental health symptoms, evidence is sparser: perseverance [19] and trait emotional intelligence [6] have consistently been found to be negatively related with depression, anxiety, and other mental health symptoms, while evidence is less robust for curiosity [20,21], creativity [22], and critical thinking [23].

**Table 1.** Overview of the Soft Skills Considered.

| Soft Skill | Definition | Theory |
|---|---|---|
| Epistemic curiosity | Drive to know | Interest/deprivation theory of curiosity (I/D, [24]) |
| Creativity | Thinking of new and effective ways to do things | Character strengths taxonomy [25] |
| Critical thinking | Analyzing and questioning learning material | Pintrich's [26] model on motivation and self-regulated learning |
| Perseverance | Prolonged effort to achieve goals | Grit [27] |
| Social awareness | Efficient interpersonal interaction | Trait emotional intelligence [28] |

### 3. Study-Related Factors and Students' Outcomes

Other than general individual features, several study-related factors have been studied in association with academic achievement, and to a lesser degree, with mental health symptoms. Following recent integrated models of academic learning [7], crucial students' intraindividual factors include emotional reactions when learning, effective cognitive and behavioral strategies (in terms of self-regulated learning [SRL] strategies), motivation to succeed, and willingness to thrive.

Achievement emotions refer to the positive and negative effects experienced with reference to the activities and/or outcomes taking place in the academic context (see control value theory; [29]. Achievement emotions have shown both direct and indirect associations with academic achievement [8,30].

SRL strategies are behaviors and thoughts students adopt to actively plan, monitor, and adjust their learning to achieve specific goals (for a review of SRL models, see [31]). These include organizing one's time, elaborating on the learning material, metacognitively reflecting upon one's learning, and so on. Evidence has shown their direct relation with academic achievement [2], as well as their mediating role in the relation between achievement emotions and academic achievement [8]. As for the relation of SRL strategies with mental health symptoms, evidence is scarce [32].

Similarly, following previous research [8,17], here we consider motivational beliefs as the interplay between academic self-efficacy (i.e., the belief in the ability to succeed in the academic context; [33]), learning goals (i.e., the individual preference for deeply mastering or performing better than others in a given learning task; [34]), and growth mindset (i.e., the belief one's intelligence is malleable and can be incremented, as opposed to being fixed; [34]). These factors have been linked to achievement both individually [2,35,36] and as a combined second-order factor [8,17]. Moreover, meta-analytical findings support a small negative association between growth mindset and psychological distress [37], and some evidence also supports the association of academic self-efficacy [38] and learning goals [39] with mental health symptoms.

Study resilience is yet another study-related factor that can have a relevant role in academic success. It is considered the ability to face particularly demanding situations and maintaining a willingness to succeed despite failures and difficulties, while being able to manage one's anxiety levels (as the general resilience construct, which is defined as successful stress-coping ability; [40]) related to study activity [41]. Study resilience was found to be positively associated with general resilience and SRL strategies, and negatively with anxiety [41].

These intraindividual factors of academic learning [7] each have a specific role in contributing to academic success in terms of academic achievement and mental health.

The build-and-broaden model [42] allows us to individuate possible variables' order. This model posits that the experience of positive emotions can enlarge the individual's momentary thought–action repertoire and ignite an upward spiraling process in which they seek out new information and experience, thus creating an enduring set of personal resources the individual can draw upon to increase their chances of successfully coping with stressful situations. In this sense, positive emotions are considered an antecedent factor able to influence the use of more functional self-regulated learning strategies and motivational beliefs, as well as building study resilience. There is evidence that single soft skills are positively related with the study-related factors considered [2,17,18,43–47], supporting the idea that they are general individual characteristics favoring the regulation of emotions, cognitions, and behaviors specifically related to learning. More specifically, curiosity and grit have been repeatedly associated with self-regulated learning, learning goals, and intrinsic motivation [17,18,45,46,48,49], while grit is associated with academic self-efficacy and achievement emotions [43,44]. Evidence is lacking for creativity and critical thinking in relation to study-related factors [47,50].

## 4. Theoretical Framework

All in all, the present study bridges three models to provide an integrative framework of the role of soft skills and study-related factors in relation to academic performance and mental health: the WEF (2015) model on personal qualities needed by XXI century students, the integrated model of SRL [7], and the broaden-and-build hypothesis [42]. To achieve such integration, the Mindsponge Theory [51] is an interesting point of reference. This conceptualization assimilates the mind to an information-processing sponge, dynamically absorbing or releasing external information based on cost–benefit judgements. The absorbed information then becomes integrated in the mindset's core belief system and changes the way in which future information is processed. In the present case, we hypothesize that more successful students (i.e., those performing better and also experiencing less distress) are those endowed with higher soft skills, in interaction with more positive achievement emotions, better SRL strategies, more functional motivational beliefs, and higher study resilience. In other words, by embedding the soft skills into their core belief system, students may gain direct benefits to their study-related processes, which in turn may make them perform and feel better.

## 5. Rationale of the Study and Hypotheses

Provided that few studies have contemporarily investigated academic achievement and mental health symptoms as relevant outcomes for university students [52], as well as the relationships between different individual features and study-related factors, the present study aims at testing a model examining the direct and indirect relations occurring between soft skills, study-related factors, and these two dependent variables. Here, we consider soft skills (i.e., epistemic curiosity, creativity, critical thinking, perseverance, and social awareness) as individual characteristics sharing a common function, i.e., regulating students' emotions, cognitions, and behaviors when learning [12,13], and, through the mediation of study-related factors, influencing academic achievement and possibly general distress. Moreover, following the broaden-and-build theory [42], we posit that experiencing

more positive achievement emotions would directly relate to the establishment of better SRL strategies, more functional motivational beliefs, and higher study resilience.

This work allows us to deepen the pattern of relationships examined in our previous work [53], in which we showed direct significant associations of soft skills and study-related factors with mental health and achievement during the first pandemic wave (2020).

Firstly, we ascertained that soft skills (i.e., epistemic curiosity, creativity, critical thinking, perseverance, and social awareness) and motivational beliefs (i.e., academic self-efficacy, growth mindset, and learning goals) are aggregated factors composed by single converging constructs [8,17,53].

Concerning the relations between soft skills, achievement emotions, motivational beliefs, SRL strategies and study resilience, and academic achievement and mental health, we hypothesized the following.

**Hypothesis 1:** *Soft skills are significantly positively associated with all four study-related factors considered (i.e., achievement emotions, SRL strategies, motivational beliefs, and study resilience; [17,18,43–48].*

**Hypothesis 2:** *Achievement emotions are significantly positively associated with SRL strategies, motivational beliefs, and study resilience [8,54–57].*

**Hypothesis 3:** *Study-related factors mediate the relation between soft skills and academic achievement [8,17]; achievement emotions and study resilience may also mediate the relation of soft skills with general distress.*

Lastly, following previous evidence, we included sex as a covariate for academic achievement [58], general distress [3], achievement emotions [59], SRL strategies [60], motivational beliefs [61,62], and study resilience [63], expecting females to show higher achievement, general distress, and SRL strategies, together with less positive motivational beliefs (academic self-efficacy), emotions, and study resilience.

## 6. Methods

Participants

A total of 606 students (153 males, *Mage* = 22.74, *SDage* = 3.72) voluntarily participated in the study. Of them, 360 students took part to the study in 2020 (96 males, *Mage* = 22.61, *SDage* = 2.88) (Data from these 360 participants of this sample were part of our previous publication [53]), while the remaining 246 students (57 males, *Mage* = 22.90, *SDage* = 4.69) participated in 2021. Participants from the two cohorts did not differ in terms of sex ($\chi^2$ = 0.77, *p* = 0.38) or age (*t* = –0.89, *p* = 0.37). The study was approved by the University of Padova's Ethics Committee for Research in Psychology (n. 3531). Table 2 shows the sociodemographic characteristics of the sample.

We performed power analysis using the pwrSEM Shiny app [64] and focused on the four indirect relations between soft skills (initial predictor), achievement emotions, SRL strategies, motivational beliefs, study resilience (mediators), academic achievement, and general distress (outcomes). Parameters were estimated to be small-to-medium, based on the previous literature presented in the above paragraphs. Power was then calculated via simulations with 10,000 iterations; results showed that with 600 participants, power was equal to 1.00 for the indirect effect of achievement emotions and SRL strategies on academic achievement, 1.00 for the indirect effect of achievement emotions and motivational beliefs on academic achievement, 1.00 for the indirect effect of achievement emotions and study resilience on academic achievement, and 1.00 for the indirect effect of achievement emotions and study resilience on general distress.

**Table 2.** Sample Characteristics.

|  | Entire Sample (*n* = 606) | Females (*n* = 453) | Males (*n* = 151) |
|---|---|---|---|
| Age | 22.74 (3.72) | 22.62 (3.67) | 23.09 (3.86) |
| Origin |  |  |  |
| Northern Italy | 464 (76.57%) | 343 (75.72%) | 121 (79.08%) |
| Central Italy | 18 (2.97%) | 15 (3.31%) | 3 (1.96%) |
| Southern Italy | 124 (20.46%) | 95 (20.97%) | 29 (18.95%) |
| Cycle |  |  |  |
| Bachelor's | 362 (59.74%) | 271 (59.82%) | 91 (59.48%) |
| Master's | 163 (26.9%) | 114 (25.17%) | 49 (32.03%) |
| Single cycle | 81 (13.37%) | 68 (15.01%) | 13 (8.5%) |
| Course year | 2.57 (1.47) | 2.52 (1.41) | 2.7 (1.62) |
| Area of study |  |  |  |
| Health sciences | 217 (35.81%) | 185 (40.84%) | 32 (20.92%) |
| Humanities | 149 (24.59%) | 123 (27.15%) | 26 (16.99%) |
| Sciences | 92 (15.18%) | 72 (15.89%) | 76 (49.67%) |
| Social sciences | 148 (24.42%) | 73 (16.11%) | 19 (12.42%) |

## 7. Materials

All the materials considered display satisfactory psychometric properties both in the original version and the present study. See Table S1 for complete information.

### 7.1. Soft Skills

**I/D Epistemic Curiosity Scale—I-Type Subscale (EC; Litman, 2008; translated in Italian by Lauriola et al., 2015 [45]).** This includes five items measuring Type I (interest) epistemic curiosity, i.e., enjoying new discoveries (e.g., "I enjoy exploring new ideas").

**Values in Action Inventory of Strengths-120—Creativity (VIA-IS; [65,66]).** This involves four items evaluating the tendency to think in new and productive ways (e.g., "Being able to come up with new and different ideas is one of my strong points").

**Motivated Strategies for Learning Questionnaire—Critical Thinking (MSLQ; [67,68]).** This involves four items examining the individual's ability to query learning material (e.g., "When a theory, interpretation, or conclusion is presented in class or in the readings, I try to decide if there is good supporting evidence").

**Short Grit Scale—Perseverance of Effort Subscale (SGS; [69,70]).** This involves four items assessing perseverance of effort, i.e., sustained effort despite setbacks (e.g., "Setbacks don't discourage me").

**Trait Emotional Intelligence Questionnaire-Short Form—Sociability Subscale (TEIQue-SF; [71,72]).** This involves six items evaluating sociability, i.e., being assertive, socially aware, effective in communication, and participatory in social situations (6 items, e.g., "I can deal effectively with people").

### 7.2. Study-Related Factors

**Emotions Questionnaire (EQ; [73]).** This involves 10 positive and 10 negative emotions experienced while studying. A total score was obtained in terms of positive academic emotions by reversing the scores for the items concerning negative emotions ($\alpha = 0.88$ in the current sample).

**Self-Regulated Learning Questionnaire—Short Form (SLQ; adapted from [41]).** This contains twenty items assessing five facets of self-regulated learning strategies (four items each): organization (e.g., "In the early afternoon I plan all the things I have to do"), elaboration (e.g., "When studying, I try to present the contents in my own words"), self-evaluation (e.g., "After a written exam, I know whether it went well or not"), preparing for exams (e.g., "I try to anticipate what kind of exam awaits me"), and metacognition (e.g., "When an exam goes wrong, I try to understand the reasons why I failed"). Only the overall score was used because it proved more reliable than the single subscales.

**Learning Goals Questionnaire (LGQ; [41]).** This includes four items on learning goals. For each item, respondents had two options to choose from, one concerning performance (e.g., "In a study situation, you prefer . . . to face tasks you already know"), and the other mastery (e.g., "In a study situation, you prefer . . . to face new tasks that you have never encountered before"). Zero points were awarded for the option representing performance goals, and one point for responses reflecting mastery goals.

**Academic Self-Efficacy Questionnaire (ASQ; [41]).** This includes five items on academic self-efficacy, i.e., the belief one can succeed in studying (e.g., "How do you rate your study skills?").

**Theories of Intelligence Questionnaire (TIQ; [41]).** This consists of eight items measuring growth mindset, i.e., the belief one's intelligence is malleable (e.g., "You can learn new things, but you can't change your intelligence"). A total score was obtained in terms of the incremental theory of intelligence by reversing the scores for the items concerning the static theory of intelligence.

**Anxiety and Resilience Questionnaire (ARQ; [41]).** It involves fourteen items investigating study anxiety (seven items, e.g., "The very thought of taking an exam makes me panic"), and study resilience (seven items, e.g., "I can overcome the disappointment over an academic failure"). An overall score was calculated, reversing the anxiety items ($\alpha = 0.86$).

*7.3. Outcome Measures*

**Grades**. Students self-reported average grades. According to the Italian university systems, grades range from a minimum of 18 to a maximum of 30.

**Depression, Anxiety, and Stress Scales-21 (DASS-21; [74,75]).** This involves twenty-one items assessing three dimensions assessed with reference to the previous week (seven items each): depression, i.e., dysphoria, low self-esteem, and lack of initiative (e.g., "I could not feel any positive emotion"); anxiety, i.e., somatic symptoms and fear responses (e.g., "I felt I was having a panic attack"); and stress, i.e., tension, high general arousal, irritability, and impatience (e.g., "I felt stressed"). A total general distress score was calculated, which proved highly reliable in the Italian validation study (Cronbach's alpha = 0.90; [75]).

*7.4. Procedure*

All participants took part in the study voluntarily and gave their consent by means of the online form before completing the self-report measures. Data were collected in 2020 (April–June) and 2021 (March–May). All the questionnaires involved in the present study were implemented in Qualtrics and took an average of 35 min to complete. A brief introduction to the study was sent to personal contacts and posted on social media, with a link to the set of questionnaires. Participants were first asked for various sociodemographic information, then they completed the questionnaires, which were presented in randomized order across participants. Lastly, they answered questions relating to their studies (e.g., average grades).

**8. Data Analysis**

RStudio [76] was used to run all the analyses.

First, following previous studies [8,11,17,77] supporting the theoretical similarity of soft skills (i.e., curiosity, creativity, perseverance, critical thinking, and social awareness) and motivational beliefs (i.e., academic self-efficacy, growth mindset, and learning goals), two confirmatory factor analyses (CFAs) with diagonally weighted least squares (DWLS) estimators were run using the package lavaan [78] to inspect these two latent variables' structure. After assessing these two latent variables' structural validity, they were converted into observed variables, as other studies have done previously [17,79].

Then, to assess the direct and indirect relations occurring between the observed variables (i.e., soft skills, achievement emotions, SRL strategies, motivational beliefs, study resilience, academic achievement, and general distress) a path model (Figure 1) was fitted. Following both theoretical considerations [42] and previous studies [8,17], the following

relationships were estimated: the direct effects of soft skills, achievement emotions, SRL strategies, motivational beliefs, and study resilience on academic achievement and general distress; the direct effects of soft skills on achievement emotions, SRL strategies, motivational beliefs, and study resilience; and the direct effect of achievement emotions on SRL strategies, motivational beliefs, and study resilience. All the indirect effects were also calculated, focusing on the indirect relations between soft skills and academic achievement through the mediation of achievement emotions, SRL strategies, motivational beliefs, and study resilience, as well as the indirect relation between soft skills and general distress through the mediation of achievement emotions and study resilience. Sex (female/male) was added as a covariate for all variables.

Lastly, multigroup confirmatory factor analysis was adopted to test model invariance across year of collection (2020/2021).

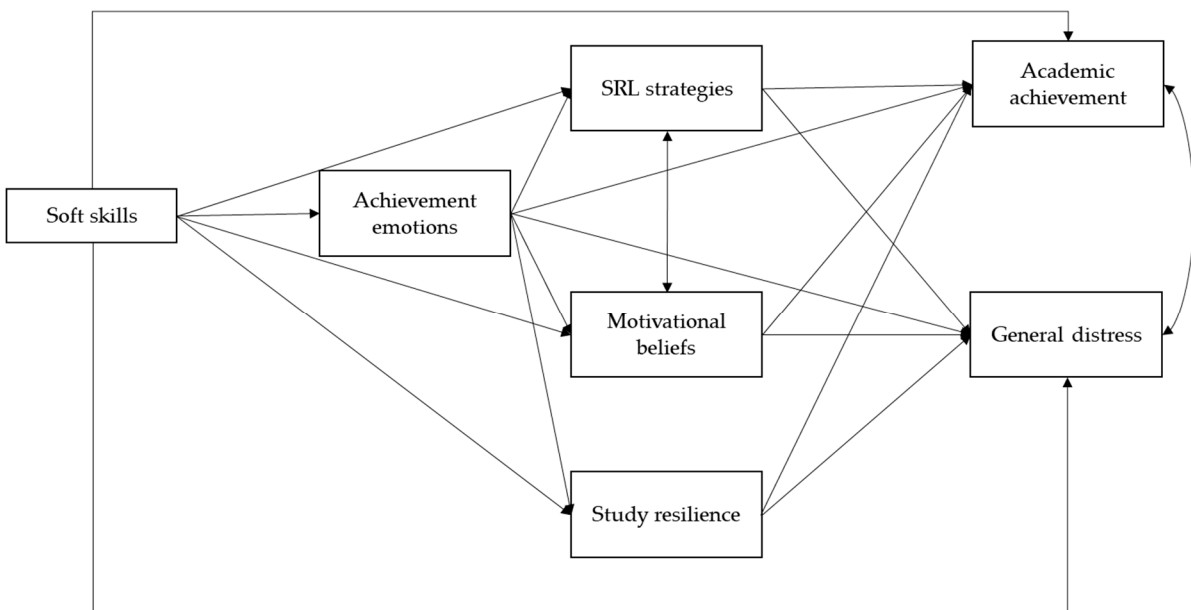

**Figure 1.** Path Model. *Note:* For Figures 1 and 2, the effect of sex is not displayed for readability.

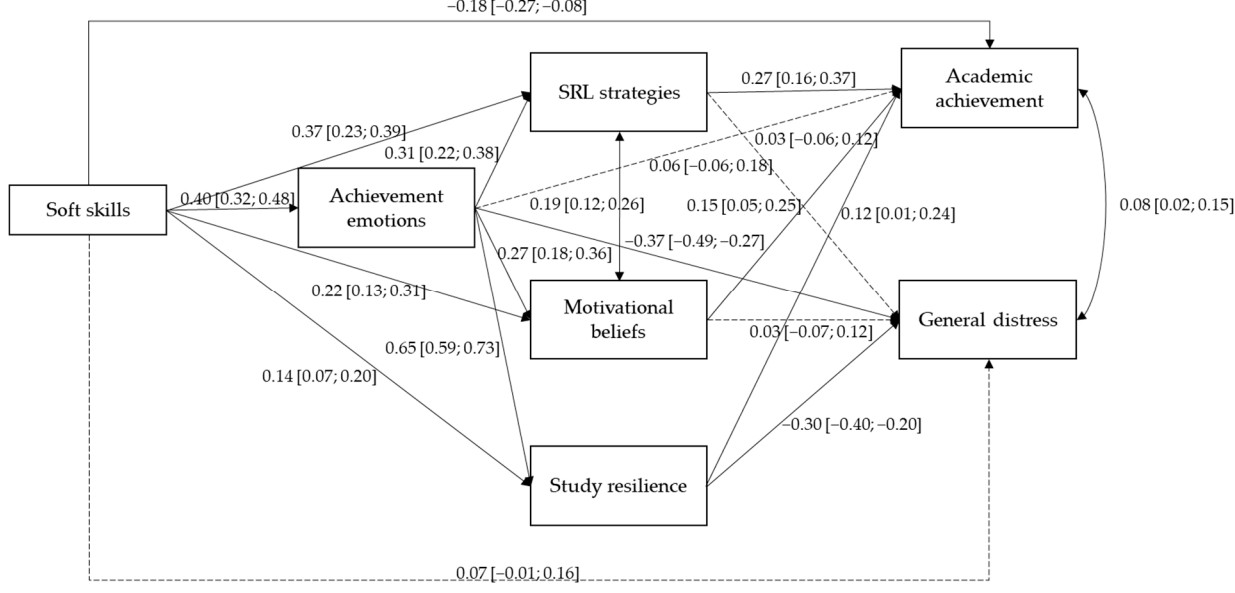

**Figure 2.** Path Model's Results.

## 9. Results

### 9.1. Factor Composition for Soft Skills and Motivational Beliefs

Two CFAs were fitted to test the structural validity of the two hypothesized latent variables.

The CFA for "soft skills" included five observed variables: epistemic curiosity (EC), creativity (VIA-IS), critical thinking (MSLQ), sociability (TEIQue), and perseverance (SGS). All the factor loadings were significant at the 0.001 level, and the average factor loading was 0.53. The fit indices were acceptable (CFI = 0.97, NNFI = 0.94, RMSEA = 0.07, SRMR = 0.05). The CFA for "motivational beliefs" included three observed variables: academic self-efficacy (ASQ), learning goals (LGQ), and theories of intelligence (TIQ). All the factor loadings were significant at the 0.001 level, and the mean factor loading was 0.46. The model was fully saturated. Full details of the CFAs' results are available in the Supplementary Materials (Table S2).

### 9.2. Path Analysis

The model showed an adequate fit to the data (CFI = 1, NNFI = 0.99, RMSEA = 0.03, 90% confidence interval for RMSEA [0.00, 0.09], SRMR = 0.01). The results mostly supported the hypothesized direct and indirect relations (Figure 2). Soft skills directly positively related with achievement emotions ($\beta = 0.40$, $p < 0.001$), SRL strategies ($\beta = 0.37$, $p < 0.001$), motivational beliefs ($\beta = 0.22$, $p < 0.001$), and study resilience ($\beta = 0.14$, $p < 0.001$), and negatively with academic achievement ($\beta = -.18$, $p < 0.001$). Achievement emotions were directly associated with study resilience ($\beta = 0.64$, $p < 0.001$), SRL strategies ($\beta = 0.31$, $p < 0.001$), and motivational beliefs ($\beta = 0.27$, $p < 0.001$). SRL strategies ($\beta = 0.27$, $p < 0.001$), motivational beliefs ($\beta = 0.15$, $p < 0.01$), and study resilience ($\beta = 0.12$, $p < 0.05$) directly related with academic achievement. Achievement emotions ($\beta = -0.37$, $p < 0.001$) and study resilience ($\beta = -0.30$, $p < 0.001$) inversely related to general distress. We also found an indirect significant relation between soft skills and academic achievement through the mediation of achievement emotions and SRL strategies ($\beta = 0.03$ [0.02; 0.05], $p < 0.001$), achievement emotions and motivational beliefs ($\beta = 0.02$ [0.01; 0.03], $p < 0.05$), and achievement emotions and study resilience ($\beta = 0.03$ [0.00; 0.06], $p < 0.05$), plus a significant indirect relation between soft skills and general distress through the achievement emotions–study resilience path ($\beta = -0.08$ [$-0.11$; $-0.05$], $p < 0.001$). The model explained 52.8% of the variance of study resilience, 35.3% of the variance of students' general distress, 34.4% of the variance of SRL strategies, 18.8% of the variance of academic achievement, 18.7% of the variance of motivational beliefs, and 16.2% of the variance of achievement emotions.

Sex, inserted as a covariate, showed significant effects in most measures considered. Compared to males, females had higher scores in academic achievement ($\beta = -0.15$ [$-0.54$; $-0.17$], $p < 0.001$), distress ($\beta = -0.09$ [$-0.37$; $-0.04$], $p < 0.05$), SRL strategies ($\beta = -0.15$ [$-0.50$; $-0.16$], $p < 0.001$), and motivational beliefs ($\beta = -0.15$ [$-0.54$; $-0.16$], $p < 0.001$), while they showed lower scores in study resilience ($\beta = 0.14$ [0.20; 0.48], $p < 0.001$). No sex effect was found for achievement emotions ($\beta = 0.05$ [$-0.06$; 0.30], $p > 0.05$).

### 9.3. Model Invariance

Since 2020 was the year in which the COVID-19 pandemic began, with decreasing effects in 2021, model invariance was calculated to ensure the model did not differ across collection year using multigroup confirmatory factor analysis to distinguish between data collected in 2020 ($n = 360$) and those collected in 2021 ($n = 246$). Results did not support scalar invariance (equality of factor loadings and intercepts) between the two cohorts ($p = 0.01$). The Lagrange multiplier test suggested that freeing the intercept for SRL strategies measures would result in an improved model. After freeing this parameter, partial scalar invariance was achieved (CFI = 0.99, NNFI = 0.97, RMSEA = 0.04, 90% confidence interval for RMSEA [0.01, 0.07], SRMR = 0.05, $p = 0.19$).

## 10. Discussion

The present study proposed a comprehensive model of several noncognitive factors as correlates of academic achievement and general psychological distress in a large sample of university students. More specifically, a model was tested in which multiple variables were considered simultaneously: five soft skills (i.e., epistemic curiosity, creativity, critical thinking, perseverance, and social awareness) were conceived as personal qualities that can affect achievement and distress through the mediation of achievement emotions, self-regulated learning, motivational beliefs (i.e., academic self-efficacy, growth mindset, and learning goals), and study resilience. This work adds to our previous analysis [53] by focusing not only on direct effects, but also on mediational effects. By considering all the variables in a single model, we could unveil the contemporary associations of intraindividual factors and our outcome variables (achievement and distress).

Results preliminarily supported the structural validity of the two latent variables hypothesized, i.e., soft skills and motivational beliefs. In other words, the present study confirms the similarity of these soft skills [11,17] and motivational factors [8,17], with the possibility to consider them as unique variables, respectively representing a series of personal qualities that can regulate emotions, behaviors, and thoughts, and functional beliefs referred to learning (as a process one can successfully manage that can be improved, and that can lead the student to competence, rather than performance only).

Then, the path model allowed us to test the direct and indirect relations linking the variables of interest. Results generally confirmed our hypotheses, showing that soft skills, as a single factor, significantly positively related with achievement emotions, SRL strategies, motivational beliefs, and study resilience. This result is in line with previous evidence on single soft skills in relation to study-related factors and further confirms the idea that these personal qualities may help students efficiently regulate their emotions, behaviors, and cognitions when studying [17,18,43–50,53]. This suggests that students who are curious, creative, and critically minded, as well as perseverant and sociable, also seem to enjoy studying more, use better strategies, feel more motivated, and are capable of recovering from failures.

Feeling more positive achievement emotions was also directly associated with the remaining study-related factors, supporting the broaden-and-build theory, according to which positive emotions favor individuals' personal resources [42]. In other words, these findings may indicate that experiencing positive emotions while studying can enlarge the set of cognitive and behavioral study strategies students use; boost their self-efficacy, growth mindset, and mastery goals; and also build their resilience. In turn, better SRL strategies, more functional motivational beliefs, and higher study resilience directly related with academic achievement, in line with past studies mainly focusing on SRL and motivation [2,8,17]. The present study additionally shows that maintaining a willingness to thrive despite academic difficulties and being able to manage one's anxiety when studying can also positively influence students' performance. This kind of reasoning is also in line with the mindsponge processes [51] of integration of new information into one's belief system. In this case, by experiencing more positive emotions, students may be willing to engage in more functional beliefs, use better study strategies, and eventually feel more resilient and less anxious in the face of study-related stressors.

As for general distress, achievement emotions and study resilience specifically resulted in significant predictors, suggesting the role of SRL strategies and motivational beliefs may be negligible when considering factors with a strong emotional component. Indeed, because distress is a general negative emotional state, experiencing positive emotions in relation to studying may have an "undoing" effect on it, buffering the impact of negative emotions students might experience in their daily lives [42]. Similarly, study resilience may protect from anxiety, depression, and stress symptoms by making students willing to appraise general stressful situations as opportunities for growth [40].

Consequently, achievement emotions, SRL strategies, motivational beliefs, and study resilience significantly mediated the relationship between soft skills and academic achieve-

ment, while only achievement emotions and study resilience were significant mediators of the relationship between soft skills and general distress.

Two other findings deserve further comments. First, contrary to our expectations and to previous meta-analytic findings [2,5,14–16], soft skills had a negative, albeit small, significant relation with academic achievement. This unexpected finding may be due to multicollinearity with the study-related factors (with which soft skills showed strong correlations), but might also indicate that soft skills alone are not sufficient to favor performance and must be considered in relation to study-related processes [17,18]. This result may also be understood in terms of a cost–benefit analysis [51], with soft skills being evaluated positively and therefore integrated in students' core systems only insofar they also benefit their study-related processes. Second, academic achievement and general distress exhibited a positive, although small, correlation. Contrary to previous findings [52] that supported a negative association between these two relevant outcomes, the present results suggest otherwise; it seems that high-performing students are also those who experience slightly higher levels of psychological distress, i.e., higher anxiety, depression, and stress symptoms. Intriguingly, studies using cluster analysis [80] have pointed out that high-achieving students may be less emotionally adjusted compared to students performing at lower levels, possibly because the former prioritize their studies over their social and emotional well-being. In this vein, it may be that students accept enduring some emotional difficulty (as expressed by higher distress) to be able to perform to higher standards, possibly because academic performance is more central to their core belief system.

Lastly, the role of sex is worth discussing. Our results showed that female students performed better than males, felt higher distress, used better SRL strategies, and held more functional motivational beliefs, but reported less study resilience. These findings are in line with previous reports [3,58,60,62,63,81] and might indicate that female students commit to their studies to a slightly higher degree compared to males and consequently may feel more distressed.

The present study has some limitations. First, results on sex differences may be biased by the higher prevalence of female participants in our sample. Similarly, the study's cross-sectional nature warrants further investigation to fully understand the direction of the present associations. Third, power analysis focused on the main five indirect paths; thus, future studies with larger samples are needed to corroborate all the direct relations evidenced here. Furthermore, future studies could adopt a Bayesian approach to overcome the multicollinearity problem, as recently outlined by Vuong et al. [82] in their Bayesian Mindsponge Framework (BMF) analytics.

Overall, the present study newly proposes to consider soft skills as general qualities jointly affecting achievement via several cognitive, behavioral, and emotional study-related processes. Soft skills seem to favor study-related factors both directly and through the mediation of positive achievement emotions. Only achievement emotions and study resilience seem to mediate the association of soft skills with general distress.

In conclusion, fostering the development and use of soft skills may have cascading positive effects for students, potentially affecting their emotional, cognitive, and behavioral learning, ultimately favoring achievement, and lowering distress.

**Supplementary Materials:** The following supporting information can be downloaded at: https://www.mdpi.com/article/10.3390/educsci13060612/s1, Table S1: Complete Information for the Measures considered in the Study; Table S2: Complete Results for the Two CFAs Assessing the Second-Order Factors, i.e., Soft Skills and Motivational Beliefs.

**Author Contributions:** Conceptualization, N.C. and C.M.; Methodology, N.C. and C.M.; Software, N.C.; Validation, N.C. and C.M.; Formal Analysis, N.C.; Investigation, N.C. and C.M.; Resources, C.M.; Data Curation, N.C.; Writing—Original Draft Preparation, N.C.; Writing—Review and Editing, C.M.; Visualization, N.C.; Supervision, C.M.; Project Administration, N.C. and C.M.; Funding Acquisition, C.M. All authors have read and agreed to the published version of the manuscript.

**Funding:** The present work was conducted as part of the Dipartimenti di Eccellenza research program (DM 11/05/2017 n. 262), supported by a grant awarded by the MIUR to the Department of General Psychology, University of Padova.

**Institutional Review Board Statement:** The study was approved by the University of Padova's Ethics Committee for Research in Psychology (n. 3531).

**Informed Consent Statement:** Informed consent was obtained from all subjects involved in the study.

**Data Availability Statement:** The data that support the findings of this study are openly available in "figshare" at http://doi.org/10.6084/m9.figshare.14822355 (accessed on 13 June 2023).

**Conflicts of Interest:** The authors have no potential interest to declare.

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
