# Peer review of "Soft Skills and Study-Related Factors: Direct and Indirect Associations with Academic Achievement and General Distress in University Students"

_education, doi:10.3390/educsci13060612_

Round 1

Reviewer 1 Report

Dear authors,
After reviewing the article I understand that it is potentially publishable with minor changes.
Overall, the article presents clear strengths:
- First, it presents a solid theoretical framework. It would not hurt to try to expand the information on the theories presented in table format and build a more integrated theoretical framework.
- Secondly, the article presents coherent objectives and appropriate hypotheses. In line with previous studies.
- Thirdly, the methodological part is presented in a correct and clear way. In this section I would like to point out the need to try to find a clearer presentation of the results. Table 3 is not entirely clear.
-Finally, I would like you to review the possibility of explaining the results more, not only to stick to the comparative part with previous studies. In the same vein, I think it would be necessary to include a section on limitations, moving lines or practical application of the results (especially the latter, given the relevant subject matter presented in the article).
I hope that these comments will be taken into account with the sole objective of improving the work,
Congratulations for the great work done,
With my best wishes

In general terms, the quality of grammar, expression and clarity in conveying the information is in line with what is expected in a scientific publication.
In this case, my first language is not English and I had no comprehension problems whatsoever.
It is true that sometimes uncommon vocabulary is used, but it is not something that reduces the quality of expression or interferes with the understanding of the article.

Author Response

Answer: Thank you very much for taking the time to read and review our work, as well as for appreciating it.

Following your insightful comments, we now provide a more integrated theoretical framework in a separate, dedicated, section: “Theoretical Framework. All in all, the present study bridges three models to provide an integrative frame-work of the role of soft skills and study-related factors in relation to academic performance and mental health: the WEF (2015) model on personal qualities needed by XXI century students, the integrated model of SRL (Ben-Eliyahu, 2019), and on the broaden-and-build hypothesis (Fredrickson, 2001). To achieve such integration, the Mind-sponge Theory (Vuong, 2023) is an interesting point of reference. This conceptualization assimilates the mind to an information-processing sponge, dynamically absorbing or releasing external information based on cost-benefit judgements. The absorbed information becomes then integrated in the mindset’s core belief system and changes the way in which future information is processed. In the present case, we hypothesize that more successful students (i.e., those performing higher and also experiencing less distress) are those endowed with higher soft skills, in interaction with more positive achievement emotions, better SRL strategies, more functional motivational beliefs and higher study resilience. In other words, by embedding the soft skills into their core belief system, students may gain direct benefits on their study-related processes, which in turn may make them perform higher and feeling better.” (see pp. 3-4).

We also decided to modify Figure 2 to include confidence intervals (see p. 9); we integrated p values and sex effects in the text (see pp. 8-9) making Table 3 unnecessary.

We expanded on limitations and implications of our results by adding the following: “This may be done by developing training interventions on all five soft skills and evaluating their specific (increase soft skills) and transfer (increase study-related factors, enhance performance, and lower distress) effects. This kind of longitudinal, randomized design may also elucidate the dynamic mental processes underlying the incorporation of the skills within students’ core beliefs (Vuong, 2023).” (see p. 13).

Please note that you will also find additional revisions at the request of the Editorial Office to distinguish this work from a previous paper already published in this journal. More specifically, we added the following in the Rationale section: “This work allows us to deepen the pattern of relationships examined in our previous work (Casali et al., 2022), in which we showed direct significant associations of soft skills and study-related factors with mental health and achievement during the first pandemic wave (2020).” (see p. 4), in the Participants section: “Of them, 360 students took part to the study in 2020 (96 males, Mage = 22.61, SDage = 2.88), while the remaining 246 students (57 males, Mage = 22.90, SDage = 4.69) participated in 2021. Participants from the two cohorts did not differ in terms of sex (χ2 = .77, p = .38) or age (t = –.89, p = .37).” (see p. 5), and in the Discussion section: “This works adds to our previous analysis (Casali et al., 2022) by focusing not only on direct effects, but also on mediational effects. By considering all the variables in a single model, we could unveil the contemporary associations of intraindividual factors and our outcome variables (achievement and distress).” (see pp. 9-10).

Reviewer 2 Report

The paper offers an interesting findings on the direct and indirect effects of soft skills on academic achievement and general distress among university students. The employed method is reliable and can be replicated, and the results are presented clearly. However, several points need to be addressed.

-        First, the Abstract needs to be rewritten to improve the clarity of results, conclusion, and implications. For example, it’s unclear what ‘four study-related factors’ are. The location of the study site is also required.

-        Second, the current study did a careful literature review of related theories and models of the studied subjects. However, those theories and models are quite fragmented, which is possible to provide a firm theoretical foundation for the proposed hypotheses. As the authors’ main objective is to explore the direct and indirect effects of soft skills, it means that they were trying to find the underlying connections between soft skills, study-related factors (achievement emotions, SRL strategies, motivational beliefs, and study resilience), academic performance, and general distress. Thus, I suggest the authors employ Mindsponge Theory to connect the presented models and theories to bolster the arguments and logic behind their hypotheses. Specifically, Mindsponge Theory helps explain how human psychology and behavior are subject to their mind’s internal information processes and exchange with the external environment. It focuses on the underlying level of human psychology and behavior through the information-processing lens but not the high levels (observations at individual and societal levels). Therefore, it does not contradict existing psychological and social theories and frameworks but rather helps elaborate, solve inconsistencies, and connect concepts through the dynamic view of information processing.

o   Vuong QH. (2023). Mindsponge Theory. De Gruyter. https://books.google.com/books?id=OSiGEAAAQBAJ

-        If you want to improve the quality of their manuscript even more by elaborating hypotheses and enriching the discussion following computational thinking, you can refer to the Bayesian Mindsponge Framework.

o   Vuong QH, Nguyen MH, La VP. (2022). The mindsponge and BMF analytics for innovative thinking in social sciences and humanities. De Gruyter. https://books.google.com/books?id=EGeEEAAAQBAJ

-        Third, the discussion is detailed but lacks depth and fails to incorporate the dynamics of the human mental process, significantly undermining the study’s values. Applying the information-processing explanatory scheme is a preferable option to remedy the shortcomings

Careful proofreading is recommended

Author Response

Answer: Thank you very much for taking the time to read and review our work and for providing such helpful comments on how to further improve it. We addressed all your comments and provided amendments in red in the revised manuscript.

-        First, the Abstract needs to be rewritten to improve the clarity of results, conclusion, and implications. For example, it’s unclear what ‘four study-related factors’ are. The location of the study site is also required.

Answer: Thank you for pointing this out. We rewrote some parts of the Abstract to make it more straightforward better presenting the results, conclusion and implications. We made it clearer that the four study-related factors are the ones mentioned in the second sentence, and we added the location of the study: “Abstract: Numerous noncognitive factors have been shown to influence students’ academic and nonacademic outcomes, yet few studies have contemporarily studied these factors to understand their specific roles. The present study tested a model in which five soft skills (i.e., epistemic curiosity, creativity, critical thinking, perseverance, and social awareness) were conceived as personal qualities that influence achievement and general distress through the mediation of four study-related factors (i.e., achievement emotions, self-regulated learning strategies, motivational beliefs, and study resilience). A total of 606 Italian university students (153 males, Mage = 22.74, SDage = 3.72) participated in the study and completed self-report measures of soft-skills, study-related factors, and general distress measures; grades were considered for academic achievement. Results showed that all four study-related factors significantly mediated the relationship of soft skills with academic achievement, while achievement emotions and study resilience only emerged as significant mediators between soft skills and general distress. Our findings indicated complex relations between individual factors and students’ outcomes due to several factors. Theoretical and practical implications are discussed.” (see p. 1)

-        Second, the current study did a careful literature review of related theories and models of the studied subjects. However, those theories and models are quite fragmented, which is possible to provide a firm theoretical foundation for the proposed hypotheses. As the authors’ main objective is to explore the direct and indirect effects of soft skills, it means that they were trying to find the underlying connections between soft skills, study-related factors (achievement emotions, SRL strategies, motivational beliefs, and study resilience), academic performance, and general distress. Thus, I suggest the authors employ Mindsponge Theory to connect the presented models and theories to bolster the arguments and logic behind their hypotheses. Specifically, Mindsponge Theory helps explain how human psychology and behavior are subject to their mind’s internal information processes and exchange with the external environment. It focuses on the underlying level of human psychology and behavior through the information-processing lens but not the high levels (observations at individual and societal levels). Therefore, it does not contradict existing psychological and social theories and frameworks but rather helps elaborate, solve inconsistencies, and connect concepts through the dynamic view of information processing.

o   Vuong QH. (2023). Mindsponge Theory. De Gruyter. https://books.google.com/books?id=OSiGEAAAQBAJ

Answer: Thank you for this insightful and helpful comment. We now integrated the suggested reference model in the manuscript by adding the following: “Theoretical Framework. All in all, the present study bridges three models to provide an integrative frame-work of the role of soft skills and study-related factors in relation to academic performance and mental health: the WEF (2015) model on personal qualities needed by XXI century students, the integrated model of SRL (Ben-Eliyahu, 2019), and on the broaden-and-build hypothesis (Fredrickson, 2001). To achieve such integration, the Mind-sponge Theory (Vuong, 2023) is an interesting point of reference. This conceptualization assimilates the mind to an information-processing sponge, dynamically absorbing or releasing external information based on cost-benefit judgements. The absorbed information becomes then integrated in the mindset’s core belief system and changes the way in which future information is processed. In the present case, we hypothesize that more successful students (i.e., those performing higher and also experiencing less distress) are those endowed with higher soft skills, in interaction with more positive achievement emotions, better SRL strategies, more functional motivational beliefs and higher study resilience. In other words, by embedding the soft skills into their core belief system, students may gain direct benefits on their study-related processes, which in turn may make them perform higher and feeling better.” (see pp. 3-4).

-        If you want to improve the quality of their manuscript even more by elaborating hypotheses and enriching the discussion following computational thinking, you can refer to the Bayesian Mindsponge Framework.

o   Vuong QH, Nguyen MH, La VP. (2022). The mindsponge and BMF analytics for innovative thinking in social sciences and humanities. De Gruyter. https://books.google.com/books?id=EGeEEAAAQBAJ

Answer: Thank you for the suggestion. We decided to integrate this line of reasoning in the Discussion section, by adding the following: “This kind of reasoning is also in line with mindsponge processes (Vuong, 2023) of integration of new information into one’s belief system. In this case, by experiencing more positive emotions, students may be willing to engage in more functional beliefs, use better study strategies, and eventually feel more resilient and less anxious in the face of study-related stressors.” (see p. 12); “This result may also be understood in terms of cost-benefit analysis (Vuong, 2023), with soft skills being evaluated positively and therefore integrated in students’ core systems only insofar they also benefit their study-related processes.” (see p. 13); “In this vein, it may be that students accept to endure some emotional difficulty (as ex-pressed by higher distress) to be able to perform to higher standards, possibly because academic performance is more central to their core belief system.” (see p. 13).

-        Third, the discussion is detailed but lacks depth and fails to incorporate the dynamics of the human mental process, significantly undermining the study’s values. Applying the information-processing explanatory scheme is a preferable option to remedy the shortcomings

Answer: Also following your previous comments, we integrated such elements by adding the following: “This kind of reasoning is also in line with mindsponge processes (Vuong, 2023) of integration of new information into one’s belief system. In this case, by experiencing more positive emotions, students may be willing to engage in more functional beliefs, use better study strategies, and eventually feel more resilient and less anxious in the face of study-related stressors.” (see p. 12); “This result may also be understood in terms of cost-benefit analysis (Vuong, 2023), with soft skills being evaluated positively and therefore integrated in students’ core systems only insofar they also benefit their study-related processes.” (see p. 13); “In this vein, it may be that students accept to endure some emotional difficulty (as ex-pressed by higher distress) to be able to perform to higher standards, possibly because academic performance is more central to their core belief system.” (see p. 13); “Furthermore, future studies could adopt a Bayesian approach to overcome the multicollinearity problem, as recently outlined by Nguyen et al. (2022) in their Bayesian Mindsponge Framework (BMF) analytics.” (see p. 13); and “This may be done by developing training interventions on all five soft skills and evaluating their specific (increase soft skills) and transfer (increase study-related factors, enhance performance, and lower distress) effects. This kind of longitudinal, randomized design may also elucidate the dynamic mental processes underlying the incorporation of the skills within students’ core beliefs (Vuong, 2023).” (see p. 13)

Please note that you will also find additional revisions at the request of the Editorial Office to distinguish this work from a previous paper already published in this journal. More specifically, we added the following in the Rationale section: “This work allows us to deepen the pattern of relationships examined in our previous work (Casali et al., 2022), in which we showed direct significant associations of soft skills and study-related factors with mental health and achievement during the first pandemic wave (2020).” (see p. 4), in the Participants section: “Of them, 360 students took part to the study in 2020 (96 males, Mage = 22.61, SDage = 2.88), while the remaining 246 students (57 males, Mage = 22.90, SDage = 4.69) participated in 2021. Participants from the two cohorts did not differ in terms of sex (χ2 = .77, p = .38) or age (t = –.89, p = .37).” (see p. 5), and in the Discussion section: “This works adds to our previous analysis (Casali et al., 2022) by focusing not only on direct effects, but also on mediational effects. By considering all the variables in a single model, we could unveil the contemporary associations of intraindividual factors and our outcome variables (achievement and distress).” (see pp. 9-10).

Round 2

Reviewer 2 Report

Although the revised manuscript doesn't satisfy me completely, especially the logical connections among arguments, I think the manuscript is qualified for publication. Also, please replace the reference 'Introduction to Bayesian Mindsponge Framework analytics: An innovative method for social and psychological research' by the book: 

Vuong QH, Nguyen MH, La VP. (2022). The mindsponge and BMF analytics for innovative thinking in social sciences and humanities. De Gruyter. https://books.google.com/books?id=EGeEEAAAQBAJ

Citing a book will help bolster the reliability of your argument more than an article.

Good luck!

Minor editing of English language required

Author Response

Thank you for your feedback. We substituted the reference to the article with the reference to the book, as you suggested.